# RECAP: Training-Free Compensation for Coarse Activation Channel Pruning in Compressed LLMs

Mingyu Lee*, Akshat Ramachandran*, Tushar Krishna

Georgia Institute of Technology, Atlanta, GA

*mlee864@gatech.edu, akshat.r@gatech.edu, tushar@ece.gatech.edu*

*Abstract*—**Sparsity is a key enabler for efficient inference in large language models (LLMs). While a wide spectrum of sparsification techniques—from unstructured to highly structured—have been explored to reduce computational overhead, they often involve trade-offs between hardware efficiency and model accuracy. Channel sparsity, in particular, is appealing due to its hardware-friendly structure compared to alternatives like structured N:M sparsity, but suffers from notable accuracy degradation, especially when applied to activations. To bridge this gap, we propose RECAP, a lightweight, training-free compensation method that mitigates the impact of channel pruning induced errors. RECAP exploits the statistics of the pruned channel as a representation of the sparsity-induced error and transfers it to the corresponding weights to compensate for the removal of the channel. Extensive experiments across diverse LLM families and benchmarks demonstrate that RECAP outperforms existing alternatives at all sparsity levels. On LLaMA3-8B, RECAP achieves approximately a 34% improvement in 0-shot BoolQ benchmark accuracy at a target sparsity ratio of 70%.**

## I. INTRODUCTION

Recently, large language models (LLMs) [3], [23] have demonstrated exceptional performance across a wide range of applications. However, their practical deployment remains challenging due to the considerable model sizes and high inference costs. To enable efficient deployment on resource-constrained devices, post-training compression techniques—including pruning and quantization—have been actively explored to reduce the computational and memory demands of serving LLMs [2], [8], [12], [19].

Model pruning [2], [24] reduces memory footprint by removing ineffectual model parameters, such as individual weights/activations (unstructured) or blocks of weights/activations (structured), and storing sparse tensors in a compressed format [7]. Most existing LLM pruning techniques [2], [8], [20], [24] primarily focus on static weight pruning. Activation sparsity, which enforces input-dependent structure on the weight matrices by leveraging (or inducing) sparsity in the input activations is relatively unexplored [12]. This can be attributed to the dynamic, input-dependent and error-prone nature of activation sparsity. Furthermore, LLMs lack non-linear functions that naturally induce sparsity unlike traditional DNNs [4].

Among existing approaches exploring activation sparsity in LLMs [11], [12], TEAL [12] proposes a simple training-free

---

*Equal contribution

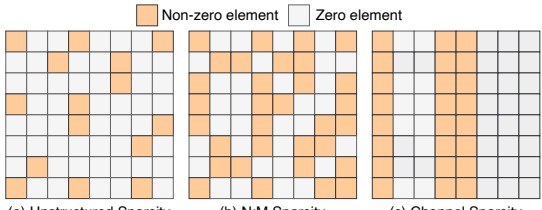

Fig. 1. Difference between (a) unstructured sparsity, (b) N:M sparsity with N=2, M=4 and (c) channel sparsity for an $8 \times 8$ representative tensor.

method that applies magnitude-based unstructured pruning across activations to achieve high model-wide sparsity, while CATS [11] targets sparsification primarily within MLP blocks by leveraging activation sparsity through gating mechanisms, but achieves lower overall sparsity due to partial layer coverage.

As demonstrated in prior work [19], [20], unstructured sparsity offers limited hardware acceleration benefits compared to more structured approaches such as N:M sparsity and channel pruning. Among these structured alternatives, channel pruning enables efficient hardware acceleration with minimal implementation complexity [9]. However, it often leads to higher compression-induced errors, posing challenges for maintaining model accuracy.

**Contributions.** Motivated by this challenge, we propose RECAP, a lightweight, training-free compensation method designed to mitigate the accuracy degradation caused by coarse-grained activation channel pruning. RECAP introduces an effective compensation strategy that leverages the average magnitude of pruned activation channels to scale the corresponding weight channels, thereby offsetting the error introduced by channel removal. To further enhance error correction in the presence of activation outliers—which can skew the average magnitude—we incorporate a fine-grained grouped channel strategy that captures localized activation distributions, leading to improved recovery fidelity. Extensive experiments across multiple LLM model families demonstrate that RECAP achieves up to 34% improvement over existing alternatives across a variety of benchmarks.

## II. BACKGROUND

### A. Sparsity in LLMs

In large language models (LLMs), sparsity primarily arises from two sources: weights and activations. Weight sparsity

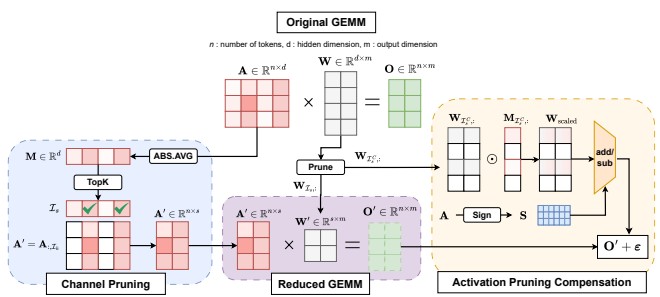

Fig. 2. Overview of RECAP. (Left) Conventional channel pruning selects channels based on magnitude, removing those with lower values. (Center) Linear Layer's GEMM operation is then reduced for efficiency. (Right) RECAP compensates the pruning error by scaling the pruned weights using the average magnitude of each channel and accumulating them based on the sign of the corresponding activation.

is typically introduced by statically eliminating redundant or ineffectual parameters, enabling model compression with minimal impact on accuracy [7], [19], [20]. In contrast, activation sparsity emerges dynamically at runtime, either as a result of non-linear operations such as ReLU [9] or by selectively pruning the outputs of inactive or low-importance neurons. The activation sparsity of a tensor $A$ is characterized by the fraction of its entries that are zero. This sparsity can influence computation through two mechanisms [12]:

- **Input sparsity:** When evaluating $O = AW$ with $A \in \mathbb{R}^{n \times d}$ and $W \in \mathbb{R}^{d \times m}$, the rows $W[i,:]$ associated with zero-valued columns in $A[:,i]$ are effectively unused.
- **Output sparsity:** Alternatively, when computing $y = s \odot (xW)$, where $s \in \mathbb{R}^{n \times m}$ is a binary mask, the rows $W[i,:]$ corresponding to entries $s[i,:] = 0$ are disregarded [10].

Following prior work [12], [20], we exploit input sparsity.

### B. Sparsification Patterns

**Unstructured Sparsity.** As illustrated in Figure 1(a), unstructured sparsity [4] eliminates individual weights or activations without adhering to any specific pattern. While it offers the highest compression ratio and minimal accuracy degradation, its irregular structure limits compatibility with hardware accelerators, thereby offering limited practical speedups [20].

**Structured N:M Sparsity.** Fine-grained N:M sparsity imposes a structured constraint by dividing weights into groups of size $M$ and retaining at most $N$ non-zero elements within each group (Figure 1(b)). While pruning remains unstructured within each group, the global enforcement of this pattern significantly improves hardware compatibility and enables efficient acceleration. This structured regularity strikes a balance between compression benefits and hardware-friendliness, making N:M sparsity a widely adopted strategy in modern sparse deep learning [7].

**Coarse-grained Channel Sparsity.** Channel sparsity treats an entire channel as an atomic unit during pruning, removing them entirely when deemed unimportant. This approach produces highly regular sparse structures, often leading to structured matrices that are well-suited for hardware acceleration [22]. However, due to its coarse-grained nature, it leads to higher accuracy degradation despite potential for superior hardware acceleration. *In this work, we aim to mitigate the*

*accuracy degradation typically associated with coarse-grained channel sparsity, thereby achieving the best of both worlds: high hardware efficiency and strong model performance.*

## III. RECAP FRAMEWORK

In this section, we introduce RECAP, *a training-free activation compensation method for coarse channel pruning in compressed large language models (LLMs)*. Building upon conventional channel pruning, RECAP exploits the statistics of the pruned channel as a representation of the sparsity-induced error and transfers it to the corresponding weights to compensate for the removal of the channel. Figure 2 provides an overview of the RECAP pipeline.

**Linear Layer GEMM Operation.** For a Linear layer in LLMs, the General Matrix Multiplication (GEMM) operation can be expressed as:

$$\mathbf{O} = \mathbf{A}\mathbf{W} \tag{1}$$

with activation $\mathbf{A} \in \mathbb{R}^{n \times d}$, weight $\mathbf{W} \in \mathbb{R}^{d \times m}$ and output $\mathbf{O} \in \mathbb{R}^{n \times m}$. Here, $n$ denotes the number of tokens, $d$ is the hidden dimension and $m$ is the output dimension. For simplicity, we assume a batch size of 1.

**Channel Pruning.** To lower the computation cost of GEMM, conventional channel pruning methods [5], [17] operates on the activation matrix $\mathbf{A}$ by first computing the average of the absolute values for each channel. A top-$k$ selection is then applied to retain only the most significant channels. This process can be formally expressed as:

$$\mathbf{M} = \frac{1}{n} \sum_{i=0}^{n-1} |\mathbf{A}_{i,:}| \tag{2}$$

$$\mathcal{I}_s = \text{TopK}(\mathbf{M}, s) \tag{3}$$

where vector $\mathbf{M} \in \mathbb{R}^d$ is the average of each channel's magnitude and $\mathcal{I}_s$ is the indices of selected top $s$ channels. Once $\mathcal{I}_s$ is obtained, its complement list $\mathcal{I}_s^C$ is used to prune the corresponding activation and weight channels as follows,

$$\mathbf{A}' = \mathbf{A}_{:,\mathcal{I}_s}, \quad \mathbf{W}' = \mathbf{W}_{\mathcal{I}_s,:} \tag{4}$$

where $\mathbf{A}' \in \mathbb{R}^{n \times s}$ and $\mathbf{W}' \in \mathbb{R}^{s \times m}$ represent the pruned activation and weight matrices, respectively, with $s << d$.

**Reduced Linear GEMM Operation.** After pruning, the GEMM computation is approximated by using only the selected top $s$ channels. The resulting operation becomes:

$$\mathbf{O} \approx \mathbf{O}' = \mathbf{A}'\mathbf{W}' = \mathbf{A}_{:,\mathcal{I}_s}\mathbf{W}_{\mathcal{I}_s,:} \tag{5}$$

This reduces the computational complexity from $O(n \cdot d \cdot m)$ to $O(n \cdot s \cdot m)$, since $s << d$, this significantly improves GEMM efficiency.

**Induced Error.** Although activations pruning is performed based on selecting the channels with highest average magnitude, it inevitably introduces an error due to the removal of channels. The reduced GEMM operation can be expressed in relation to the original GEMM as follows:

$$\mathbf{A}\mathbf{W} = \mathbf{A}'\mathbf{W}' + \varepsilon + \delta \tag{6}$$

where $\varepsilon + \delta$ denotes the total error resulting from channel pruning. In this formulation, $\delta$ refers to the irrecoverable error, while $\varepsilon$ denotes the recoverable portion. The objective of RECAP is to maximize compensation for the recoverable error $\varepsilon$, thereby mitigating the impact of the irrecoverable error $\delta$.

**RECAP's Compensation.** RECAP compensates $\varepsilon$ by leveraging the precomputed statistics of the pruned channel which is the average of each channel's magnitude, $\mathbf{M}_{\mathcal{I}_s^C}$. With vector $\mathbf{M}_{\mathcal{I}_s^C}$, we scale the corresponding pruned rows of the weight matrix through element-wise multiplication as:

$$\mathbf{W}_{\text{scaled}} = \mathbf{M}_{\mathcal{I}_s^C} \odot \mathbf{W}_{\mathcal{I}_s^C,:} \tag{7}$$

Here, $\odot$ represents element-wise multiplication, with broadcasting of the vector $\mathbf{M}_{\mathcal{I}_k^C}$ over to the rows of $\mathbf{W}_{\mathcal{I}_k^C,:}$. Simultaneously, sign bits of the activation matrix $\mathbf{A}$ are extracted to guide the compensation process. These sign values determine whether each scaled weight is added or subtracted. It primarily functions to prevent nullification of the compensation. The compensated error $\varepsilon$ is then computed by accumulating the scaled weights accordingly:

$$\mathbf{S} = \text{sign}(\mathbf{A}) \tag{8}$$

$$\varepsilon = \sum_{i=0}^{n-1} \sum_{j=0}^{m-1} \sum_{k \in \mathcal{I}_s^C} \mathbf{S}_{i,k} \cdot \mathbf{W}_{\text{scaled}}^{(k,j)} \tag{9}$$

where $\varepsilon \in \mathbb{R}^{n \times m}$ is the compensation tensor and $\mathbf{W}_{\text{scaled}}^{(k,j)}$ denotes the $(k,j)$ element of the scaled weight matrix.

**Fine-grained Compensation.** To better approximate $\varepsilon$, we propose a fine-grained grouped channel compensation strategy. This approach improves the accuracy of restoration by computing the average magnitude $\mathbf{M}$ at a finer granularity, thereby enabling better approximation even under skewed channel value distributions. This skewed channel distribution typically arises from the varied distribution of inliers and outliers in channels.

In this strategy, each channel to be compensated is divided into groups of size $\mathbf{N}_g$ along the token dimension, and per-group statistics are computed within the pruned channel subset. The formulation is as follows:

$$\mathbf{n}_g = n / \mathbf{N}_g \tag{10}$$

$$\mathbf{M}_g = \frac{1}{\mathbf{n}_g} \sum_{i=\mathbf{n}_g \cdot g}^{\mathbf{n}_g \cdot g + \mathbf{n}_g - 1} |\mathbf{A}_{i,:}| \tag{11}$$

where $\mathbf{M}_g$ denotes the average magnitude for group $g$ within the channels.

$$\mathbf{W}_{\text{scaled}}^{(g)} = \mathbf{M}_{\mathcal{I}_s^C}^{(g)} \odot \mathbf{W}_{\mathcal{I}_s^C,:} \tag{12}$$

$$\varepsilon_g = \sum_{i=\mathbf{n}_g \cdot g}^{\mathbf{n}_g \cdot g + \mathbf{n}_g - 1} \sum_{j=0}^{m-1} \sum_{k \in \mathcal{I}_s^C} \mathbf{S}_{i,k} \cdot \mathbf{W}_{\text{scaled}}^{(g,k,j)} \tag{13}$$

$$\varepsilon = \text{concat}(\varepsilon_0, \varepsilon_1, ..., \varepsilon_{\mathbf{N}_g - 1}) \tag{14}$$

TABLE I
COMPARISON OF UNIFORM PRUNING ON PERPLEXITY, ZERO SHOT BOOLQ, 5 SHOT MMLU FOR LLaMA2-7B [23]/LLaMA3-8B [3]/QWEN2.5-3B [18]. THE RECAP GROUP SIZE IS SET TO $\mathbf{N}_g = 8$.

| Model | Sparsity | Method | PPL(↓) | BoolQ(↑) | MMLU(↑) |
|---|---|---|---|---|---|
| | Baseline | - | 5.47 | 77.74 | 45.81 |
| | 30% | Channel Pruning | 57.95 | 62.29 | 25.42 |
| | | **RECAP (Ours)** | 9.80 | 72.72 | 32.07 |
| LLaMA2-7B [23] | 50% | Channel Pruning | $1.65 \cdot 10^4$ | 48.10 | 23.87 |
| | | **RECAP (Ours)** | 16.69 | 69.94 | 26.82 |
| | 70% | Channel Pruning | $9.25 \cdot 10^3$ | 43.70 | 23.72 |
| | | **RECAP (Ours)** | 72.30 | 67.09 | 25.17 |
| | Baseline | - | 6.24 | 82.08 | 65.41 |
| | 30% | Channel Pruning | 192.51 | 62.20 | 26.19 |
| | | **RECAP (Ours)** | 29.32 | 78.44 | 40.50 |
| LLaMA3-8B [3] | 50% | Channel Pruning | $8.50 \cdot 10^3$ | 42.32 | 26.04 |
| | | **RECAP (Ours)** | 33.46 | 78.35 | 39.25 |
| | 70% | Channel Pruning | $3.85 \cdot 10^4$ | 38.75 | 24.72 |
| | | **RECAP (Ours)** | 94.29 | 72.48 | 27.91 |
| | Baseline | - | 8.03 | 77.10 | 65.61 |
| | 30% | Channel Pruning | 50.36 | 61.23 | 26.24 |
| | | **RECAP (Ours)** | 6.43 | 74.36 | 49.51 |
| Qwen2.5-3B [18] | 50% | Channel Pruning | $1.26 \cdot 10^3$ | 53.37 | 25.17 |
| | | **RECAP (Ours)** | 10.36 | 70.26 | 47.85 |
| | 70% | Channel Pruning | $3.19 \cdot 10^4$ | 47.31 | 23.58 |
| | | **RECAP (Ours)** | 37.35 | 66.16 | 33.65 |

---

*(50% Pruning Level, LLaMA2-7B)*
**Input Text** : The largest animal
**Baseline** : is the blue whale. It can grow to 30m in length and weigh over ..
**Channel** : is the elephant, which weighs up to 6 tons and can reach 25 feet..
**RECAP** : is the blue whale, which can reach 30 meters (98 ft) in length and..

**Input Text** : A Large Language Model is
**Baseline** : a type of deep learning model that has been trained on a ..
**Channel** : a language that is used by a large number of people..
**RECAP** : a type of artificial intelligence (AI) model that is capable of..

Fig. 3. Text generation examples from LLaMA2-7B under 50% sparsity ratio. Incorrect outputs and correct outputs are highlighted.

Here, $\mathbf{W}_{\text{scaled}}^{(g)}$ represents the scaled weights for group $g$. The term $\varepsilon_g \in \mathbb{R}^{\mathbf{n}_g \times m}$ corresponds to the compensation tensor for group $g$, aggregated across spatial and channel dimensions. Finally, the complete compensation tensor $\varepsilon$ is formed by concatenating the group-wise compensation tensors across all $\mathbf{N}_g$ groups.

RECAP transfers the statistical information of pruned channels to their corresponding weights, transforming full matrix multiplications into lightweight operations comprising element-wise scaling and sign-guided accumulation. This design enables effective $\varepsilon$ compensation with minimal computational overhead.

## IV. EXPERIMENTAL EVALUATIONS

### A. Experimental Setup

**Models and Datasets.** We evaluate RECAP on LLM families LLaMA2-7B [23], LLaMA3-8B [3] and Qwen2.5-3B [18]. Each model's perplexity is measured with WikiText2 [16] dataset and benchmark accuracy on BoolQ [1], MMLU [6] and HellaSwag [25] (ablations), under both conventional channel sparsity without compensation and the proposed RECAP compensation.

**Implementation Details.** RECAP is implemented in PyTorch and all experiments are conducted on a single NVIDIA GH200 GPU. We set the group size as $\mathbf{N_g} = 8$, which is empirically determined. RECAP is applied to activations in linear layers,

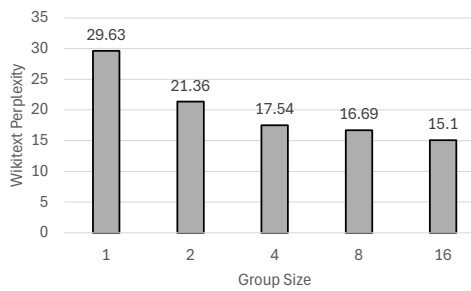

Fig. 4. Effect of group size ($\mathbf{N_g}$) on Wikitext perplexity for LLaMA2-7B [23] at uniform sparsity of 50%.

TABLE II
NON-UNIFORM LAYER-WISE PRUNING COMPARISON BETWEEN
TEAL [12] AND RECAP FOR LLAMA3-8B [3] AT A TARGET SPARSITY
RATIO OF 60%.

| Method | BoolQ (↑) | HellaSwag (↑) | Average (↑) |
|---|---|---|---|
| Baseline | 82.08 | 82.20 | 82.14 |
| TEAL [12] | 73.26 | 69.85 | 71.56 |
| **RECAP (Ours)** | 78.36 | 74.63 | 76.50 |

targeting query, key, value, attention output projections and feed-forward layers inside the transformer block.

### B. Perplexity Evaluation

In Table I, we compare the perplexity results of Wiki-Text2 on LLaMA2-7B, LLaMA3-8B and Qwen2.5-3B across 30%/50%/70% uniform pruning. RECAP consistently achieves substantially lower perplexity compared to channel pruning across all sparsity ratio and models, demonstrating it's effectiveness in recovering the error $\varepsilon$. Notably, under 50% sparsity ratio, RECAP reduces the perplexity by up to $3.84 \cdot 10^4$, showing its robustness in maintaining model performance even under aggressive pruning.

### C. Downstream Task Evaluation

As presented in Table I, RECAP consistently recovers a substantial portion of the lost accuracy from channel pruning across BoolQ and MMLU benchmarks, highlighting its robustness across a variety of tasks and architectures. Notably, RECAP restores up to 34% accuracy on LLaMA3-8B for BoolQ and 23% accuracy on Qwen2.5-3B for MMLU compared to conventional channel pruning.

### D. Text Generation

In Figure 3, we compare the generated sentences with naive channel pruning and RECAP with Llama2-7B model under 50% activation pruning. Compared to channel pruning without compensation, RECAP significantly improves output quality, effectively mitigating semantic distortions caused by pruning. For instance, when asked about the largest animal, channel pruning produces an incorrect output referring to elephants, whereas RECAP correctly identifies the blue whale, demonstrating its ability to accurately recover lost information.

### E. Ablations and Discussions

**Effect of group size $\mathbf{N}_g$.** We illustrate the impact of varying $\mathbf{N}_g$ values on model perplexity in Figure 4. As observed, increasing $\mathbf{N}_g$ consistently leads to lower perplexity, attributed to improved error approximation at finer granularity, which better captures skewed channel distributions caused by outliers. However, setting $\mathbf{N}_g$ too large degrades inference performance due to increased computational overhead. To balance accuracy and efficiency tradeoff, we select $\mathbf{N}_g = 8$.

**Non-uniform Layer-wise Pruning.** Since TEAL [12] applies non-uniform layer-wise pruning using a greedy optimization, we employ FLOW [20] to determine optimal layerwise sparsity ratios for RECAP, ensuring a fair comparison at a target sparsity of 60%. Both methods use a calibration set of 256 samples drawn from WikiText [16]. As shown in Table II, RECAP achieves approximately 5% higher benchmark accuracy than TEAL, highlighting the effectiveness of RECAP's compensation for pruning-induced errors.

**Future Directions.** Our technique is orthogonal to existing weight compression methods and can be integrated alongside them to enable further model compression, which we plan to investigate this integration in future work. Additionally, we plan to theoretically validate our method and also develop a formulation for the unrecoverable error component $\delta$.

## V. RELATED WORK

**LLM Pruning.** Recent advances in unstructured pruning, such as SparseGPT [2], leveraged the inverse Hessian for importance estimation, while Wanda [21] proposed a simple criterion based on the product of weight magnitudes and activations. Extensions of these techniques to structured N:M sparsity have largely been restricted to applying a fixed N:M pattern uniformly across all layers. However, works such as [20], [24] emphasized the need for heterogeneous sparsity budgets across different layers, advocating for a non-uniform, layer-wise N:M sparsity assignment based on the presence and distribution of outliers. In parallel, recent research [12] explored magnitude pruning of LLM activations, motivated by the observation that activation distributions are typically zero-mean and unimodal.

**Error Compensation.** To compensate for the sparsity induced errors, early techniques as [15] are not practical for large LLMs due to the demand for significant compute resources required for model finetuning. To overcome this challenge, techniques such as [2] leverage the inverse Hessian to compensate for sparsification errors, [14] explored parameter-efficient finetuning for sparsified LLMs, [13] projects compression errors into a low-rank space, and minimizes compression-induced errors without requiring gradient-based training.

## VI. CONCLUSION

We propose RECAP, a lightweight, training-free compensation method to mitigate the impact of channel pruning errors. Utilizing the statistics of pruned activation channels, RECAP transfers the approximation to corresponding weights and effectively compensates pruning induced errors. Extensive experiments across various LLM families demonstrate that RECAP achieve substantial error recovery, consistently improving benchmark accuracy and perplexity. Our approach

highlights the potential of RECAP to bridge the gap between hardware efficiency and model performance in the context of pruning, offering a promising future direction of exploration.

## ACKNOWLEDGMENTS

This work was supported in part by CoCoSys, one of the seven centers in JUMP 2.0, a Semiconductor Research Corporation (SRC) program sponsored by DARPA.

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
