# OpenReview forum: "RECAP: Training-Free Compensation for Coarse Activation Channel Pruning in Compressed LLMs"
_iscaconf.org/ISCA/2025/Workshop/MLArchSys — MLArchSys 2025 Oral_

### Official Review · Reviewer_Wn7k · 2025-05-16
**Practical and Effective Method, Though Theoretical Depth is Limited**

**Confidence:** 4
**Rating:** 6

**Detailed Feedback And Questions For Authors:**

The paper addresses a relevant challenge in the LLM compression space: how to mitigate the quality degradation introduced by coarse channel pruning of activations, which is structurally favorable for hardware but detrimental for accuracy. The proposed method, RECAP, is elegant and practical. It avoids retraining by using activation magnitude statistics and sign-aware accumulation to compensate pruning-induced errors at inference time.

The results are compelling: RECAP significantly improves both perplexity and task accuracy across three LLMs (LLaMA2, LLaMA3, Qwen2.5) and several benchmarks (WikiText, BoolQ, MMLU). The gains are particularly pronounced at high sparsity levels (50%–70%), suggesting the method is robust under aggressive pruning.

However, the paper would benefit from additional theoretical grounding. For instance, the decomposition into ε (recoverable error) and δ (unrecoverable error) is intuitive but lacks analytical justification or bounds. It would strengthen the work to show when and why RECAP’s compensation aligns well with true residuals. Moreover, comparisons to related training-free methods like EORA or SPP are only briefly touched upon.

Ablation on group size and sample qualitative generations are helpful, though a breakdown by layer or component (e.g., attention vs. MLP) would better illuminate where compensation is most needed or effective. The method’s orthogonality to weight pruning is mentioned but not explored.

In summary, this paper offers a practical contribution with solid empirical support. While not a breakthrough in theory, it fills an important gap between compression efficiency and quality retention in LLMs, justifying acceptance.

**Top Reasons To Accept The Paper:**

Proposes a lightweight, training-free method (RECAP) to compensate for accuracy loss due to activation channel pruning, a relatively underexplored direction in LLM compression.

Demonstrates consistent empirical gains across multiple models and benchmarks, including perplexity, BoolQ, and MMLU, with up to 34% improvement over pruning baselines.

The compensation strategy is simple yet effective, and does not require model retraining or fine-tuning, which is valuable for practical deployment.

**Top Reasons To Reject The Paper:**

Theoretical insights are lacking—the formulation of the unrecoverable error and the justification for compensation strategy are mostly empirical.

Evaluation focuses heavily on activation channel pruning without exploring broader applicability to other sparsity formats or integration with quantization.

---

### Official Review · Reviewer_9CCg · 2025-05-17
**Simple yet effective sparsity error correction method - overheads remain unclear**

**Confidence:** 4
**Rating:** 7

**Detailed Feedback And Questions For Authors:**

Quality:
The paper effectively demonstrates the quality impact of its method across various models and tasks. It's well-written, and the proposed method is clearly laid out.

Clarity:
The paper is well-written, guiding the reader smoothly through the method and effectively contextualizing the work within the existing literature.

Originality:
While the formulation of the correction term draws inspiration from previous ideas, its specific form in this context is novel.

Significance:
The paper clearly demonstrates a quality advantage over other methods. However, it does not present any speed-up numbers. Despite this, the method's simplicity and effectiveness suggest it could have a significant impact.

**Top Reasons To Accept The Paper:**

Novel Pruning Correction: The method introduces a correction term for pruning error based on activation average magnitude and weights, which is a relatively simple yet advantageous formulation compared to other works.

Quality Results: The approach yields positive results across all benchmarks, outperforming pure channel pruning methods.

Breadth of Benchmarks: The authors rigorously demonstrate the quality benefits of their method through evaluations on various models, perplexity metrics, and downstream tasks.

Clarity in Writing: The paper is clearly written, making the concepts easy to understand.

**Top Reasons To Reject The Paper:**

Unclear Overheads: The paper lacks clarity regarding the overheads of the sparsification method. Specifically, it's not clear whether sparsification occurs at runtime or is fixed based on a calibration dataset.
+ If at runtime, a discussion on the computational expense of applying the method versus the gains from accelerated matrix multiplications would be beneficial.
+ If performed offline with a calibration dataset, the paper should address how the choice and characteristics of the calibration dataset impact the resulting quality.

Limited Comparisons: The paper could have benefited from comparisons against a broader range of existing works (the pure channel-wise benchmark seems to be weak benchmark, but it is also understood that other methods might have different overheads making direct comparisons non-trivial).

Quality-Performance Trade-off: The method does not fully recover baseline accuracies, indicating that it presents a quality-performance trade-off.

---

### Official Review · Reviewer_qJhh · 2025-05-18
**Review summary**

**Confidence:** 5
**Rating:** 5

**Detailed Feedback And Questions For Authors:**

This paper introduces RECAP, a training-free quality compensation method for LLM activation channel pruning. The main goal is to leverage the statistics of the pruned channels (average magnitude) to scale the corresponding weights, resulting in additional memory and compute added in return to compensate for the error induced by the channel removal. Results show RECAP can improve LLM serving quality when activation channel pruning is enabled.

- Pros:

The paper clearly identifies the challenges of activation sparsity in LLMs. It is easy to follow and well-organized within the page limit.

- Cons:

1. The overhead of RECAP is not well explained.  With RECAP, the weight matrix is scaled up from the pruned shape to the original shape and additional ops are needed. It will be more clear to showcase the quality-efficiency tradeoff by adding efficiency metrics, such as inference latency.
2. The distinction between recoverable (ε) and irrecoverable (δ) error could be further elaborated. Providing insights into what constitutes each type of error and how RECAP specifically targets the recoverable portion would strengthen the explanation.
3. RECAP is applied to multiple linear layers. It is unclear if the same RECAP setup is applied to all these layers or it needs to be configured differently.

**Top Reasons To Accept The Paper:**

None

**Top Reasons To Reject The Paper:**

None

---

### Official Review · Reviewer_wk22 · 2025-05-18
**Quality recovery for channel pruning**

**Confidence:** 4
**Rating:** 6

**Detailed Feedback And Questions For Authors:**

The paper provides a simple and effective idea to recover the quality lost due to channel pruning. It provides the necessary evaluations to test the idea and demonstrate the results. To improve, it should include more rigorous comparative study against existing work and other alternative approaches.

**Top Reasons To Accept The Paper:**

- Improves over naive channel pruning
- Provides evaluations using SOTA LLMs
- Compares against a relevant existing work

**Top Reasons To Reject The Paper:**

- Would be good to compare against N:M fine-grained sparsity techniques, to see where channel pruning stands.
- Limited comparative analysis against the existing work.
- Even though it recovers quality a lot, the end results show heavy degradation when compared to the dense baseline.